# Cryo-EM structures of the BAF-Lamin A/C complex bound to nucleosomes

Naoki Horikoshi[1,2], Ryosuke Miyake[1,3], Chizuru Sogawa-Fujiwara [1], Mitsuo Ogasawara[1], Yoshimasa Takizawa [1,4] & Hitoshi Kurumizaka [1,3,5]✉

Barrier-to-autointegration factor (BAF) associates with mitotic chromosomes and promotes nuclear envelope assembly by recruiting proteins, such as Lamins, required for the reconstruction of the nuclear envelope and lamina. BAF also mediates chromatin anchoring to the nuclear lamina via Lamin A/C. However, the mechanism by which BAF and Lamin A/C bind chromatin and affect the chromatin organization remains elusive. Here we report the cryo-electron microscopy structures of BAF-Lamin A/C-nucleosome complexes. We find that the BAF dimer complexed with the Lamin A/C IgF domain occupies the nucleosomal dyad position, forming a tripartite nucleosomal DNA binding structure. We also show that the Lamin A/C Lys486 and His506 residues, which are reportedly mutated in lipodystrophy patients, directly contact the DNA at the nucleosomal dyad. Excess BAF-Lamin A/C complexes symmetrically bind other nucleosomal DNA sites and connect two BAF-Lamin A/C-nucleosome complexes. Although the linker histone H1 competes with BAF-Lamin A/C binding at the nucleosomal dyad region, the two BAF-Lamin A/C molecules still bridge two nucleosomes. These findings provide insights into the mechanism by which BAF, Lamin A/C, and/or histone H1 bind nucleosomes and influence chromatin organization within the nucleus.

In eukaryotes, genomic DNA is bound to various nuclear proteins and compacted as chromatin in the nucleus. The fundamental repeating unit of chromatin is the nucleosome, which contains two copies of histones H2A, H2B, H3, and H4[1]. In chromatin, nucleosomes are connected by linker DNAs and appear as fibers with a beads-on-a-string configuration[2,3]. Higher-order chromatin compaction and its dynamics, which are epigenetically regulated, are crucial for DNA replication, transcription, recombination, and repair[4–6]. The three-dimensional arrangements and dynamics of chromatin architecture in the nucleus are spatiotemporally regulated for proper development and cell differentiation[7].

In the interphase of the cell cycle, genomic DNA is spread out within the nucleus as chromatin fibers. The chromatin at the nuclear periphery is compacted and anchored to the nuclear lamina, a fibrous meshwork comprising Lamins, inner nuclear membrane proteins, and nuclear lamina-associated proteins[8,9]. The interactions between chromatin and the nuclear lamina at the nuclear periphery play pivotal roles in chromatin conformation and genome regulation. In contrast, in the M-phase, the nuclear envelope breaks down, and genomic DNA is highly compacted as mitotic chromosomes by the condensation of chromatin fibers. Mitotic chromosomes are aligned at the center of the cell, and each pair of sister chromatids is subsequently segregated into daughter cells. Following chromosome segregation, the nuclear envelope is reassembled to enclose the gathered chromosomes[10–12].

Barrier-to-autointegration factor (BAF) associates with chromatin and plays a crucial role in priming the nuclear envelope

[1]Laboratory of Chromatin Structure and Function, Institute for Quantitative Biosciences, The University of Tokyo, 1-1-1 Yayoi, Bunkyo-ku, Tokyo, Japan. [2]Department of Cell Biology and Anatomy, Graduate School of Medicine, The University of Tokyo, 1-1-1 Yayoi, Bunkyo-ku, Tokyo, Japan. [3]Department of Biological Sciences, Graduate School of Science, The University of Tokyo, 1-1-1 Yayoi, Bunkyo-ku, Tokyo, Japan. [4]Department of Computational Biology and Medical Sciences, Graduate School of Frontier Sciences, The University of Tokyo, 1-1-1 Yayoi, Bunkyo-ku, Tokyo, Japan. [5]Laboratory for Transcription Structural Biology, RIKEN Center for Biosystems Dynamics Research, 1-7-22 Suehiro-cho, Tsurumi-ku, Yokohama, Japan. ✉e-mail: kurumizaka@iqb.u-tokyo.ac.jp

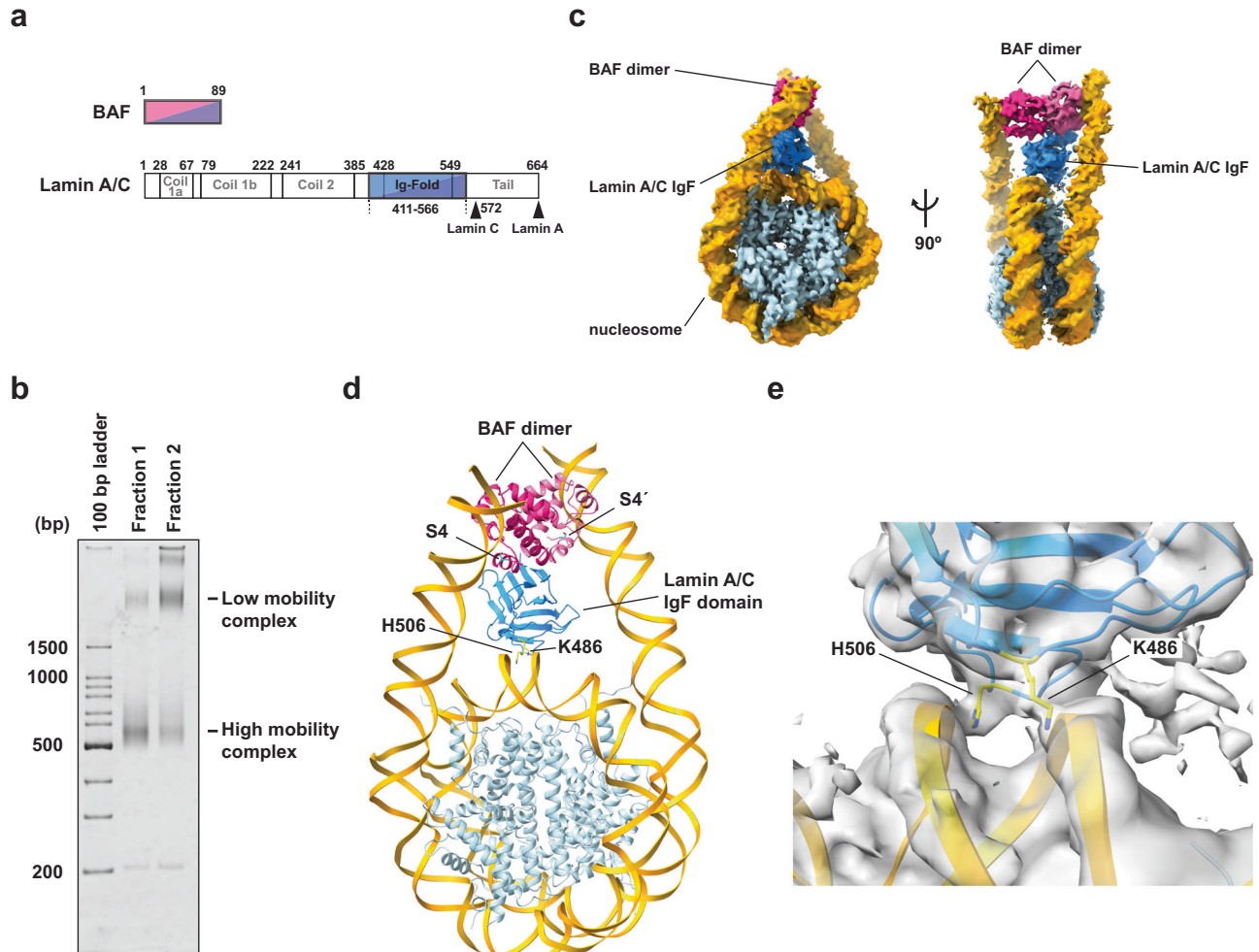

**Fig. 1 | Cryo-EM structure of the BAF-Lamin A/C IgF-nucleosome complex.**
**a** Schematic representation of the domain compositions of BAF and Lamin A/C. Full-length BAF and the Lamin A/C IgF domain used in this study are colored pink/purple and sky blue/deep blue, respectively. **b** The BAF-Lamin A/C IgF-nucleosome complexes prepared by the GraFix method were analyzed by native-polyacrylamide gel electrophoresis (PAGE) with ethidium bromide staining. This complex has been independently prepared three times using the GraFix method, yielding consistent results. **c** Cryo-EM map of the BAF-Lamin A/C IgF-nucleosome complex in fraction 1 of the panel (**b**). Histones, DNA, BAF, and Lamin A/C IgF are colored light blue, gold, pink, and blue, respectively. **d** Structure of the BAF-Lamin A/C IgF-nucleosome complex. **e** Close-up view of the interface between the nucleosomal dyad and the Lamin A/C IgF domain. The model is overlaid with the Cryo-EM map, colored gray. The Lamin A/C K486 and H506 residues are colored yellow. Source data are provided as a Source Data file.

reassembly around the mitotic chromosomes[13]. In late anaphase and early telophase during mitosis, BAF accumulates at the "core region" of chromosomes, where nuclear lamina proteins are subsequently recruited and promote nuclear envelope reassembly[13–15]. During the nuclear envelope reassembly processes, BAF binds to chromatin and connects Lamin A/C, a nuclear lamina protein, to chromatin by interacting with its Immunoglobulin-fold (IgF) domain[15–19]. Consistently, the depletion of BAF leads to nuclear membrane abnormalities, multinucleation, and micronuclei formation[20–22]. Loss of BAF results in mitotic defects and embryonic lethality in *C. elegans* and *D. melanogaster*[20,23,24]. A mutation in human BAF causes Nestor-Guillermo Progeria Syndrome (NGPS), with accompanying abnormalities of the nuclear envelope and higher-order chromatin conformations[18,25–27]. In interphase cells, BAF functions to anchor chromatin to the nuclear lamina by the BAF-Lamin A/C interactions[16,18,26]. However, the mechanism by which BAF binds chromatin with Lamin A/C has remained elusive.

In the present study, we show the cryo-electron microscopy (cryo-EM) structures of BAF-Lamin A/C IgF-nucleosome complexes with or without linker histone H1. These structures provide the structural basis of chromatin binding by BAF and Lamin A/C in eukaryotic cells.

## Results
### BAF-Lamin A/C binds the nucleosome by tripartite DNA binding
To analyze the mechanism by which BAF binds to chromatin and the nuclear lamina, we reconstituted the nucleosome complexed with BAF and the Lamin A/C IgF domain, which is the intranuclear BAF-binding domain of nuclear lamina (Fig. 1a). To do so, we prepared the nucleosome with 24 base-pair linker DNAs at both ends (Supplementary Fig. 1a, b). Purified full-length BAF and the Lamin A/C IgF domain were incubated with the nucleosome (Supplementary Fig. 1c), and the resulting complexes were separated by sucrose-gradient ultracentrifugation in the presence of glutaraldehyde (GraFix). We obtained two distinct complexes, with high and low mobilities on a non-denaturing polyacrylamide gel (Fig. 1b).

These high and low mobility complexes were then separated by sucrose gradient ultracentrifugation with glutaraldehyde fixation (GraFix) (Supplementary Fig. 1f). Two complexes were also observed in sucrose gradient ultracentrifugation without glutaraldehyde, although the ratio of the low mobility complex was reduced (Supplementary Fig. 1e, f). Therefore, these two complexes, with high and low mobilities, are not artificially formed due to cross-linking. We then conducted the cryo-EM single particle analysis with the high gel mobility

complex fraction and determined the structure of the BAF-Lamin A/C-nucleosome complex (Fig. 1b, cand Supplementary Fig. 2a). The BAF dimer and the Lamin A/C IgF domain interact in this complex, as previously reported[16]. We found that BAF-Lamin A/C IgF is located on the dyad region of the nucleosome, and each BAF protomer binds a linker DNA, tying two linker DNAs together in the nucleosome (Fig. 1c, d). The two linker DNA fragments bound to the BAF dimer in the nucleosome are perpendicularly aligned, consistent with the orientation of the two DNA fragments in the BAF-DNA complex reported previously (Supplementary Fig. 3a)[28]. The side chain moieties of the Lamin A/C Lys486 and His506 residues directly contact the DNA at the nucleosomal dyad (Fig. 1e). Consequently, the BAF-Lamin A/C IgF complex binds nucleosomal DNA in a tripartite manner (Fig. 1c, d).

### Mutational analysis of nucleosome binding by BAF and Lamin A/C

To test the nucleosome binding of BAF and Lamin A/C, we performed a nucleosome pull-down assay in which BAF and Lamin A/C bound to bead-conjugated nucleosomes were detected by SDS-polyacrylamide gel electrophoresis (SDS-PAGE) (Fig. 2a). We found that the Lamin A/C IgF domain does not bind to the nucleosome without BAF, although BAF binds to the nucleosome without the Lamin A/C IgF domain (Fig. 2b, e, lanes 7 and 8). Therefore, the nucleosome binding of the Lamin A/C IgF domain requires the BAF dimer. The Lamin A/C K486N and H506D mutations, in which the Lys486 and His506 residues are replaced by Asn and Asp (K486N and H506D), respectively, have been identified in lipodystrophy patients[29–31]. A previous study reported that the nuclear organization of chromatin domains is substantially disordered in cells isolated from these patients[32]. We then tested the nucleosome binding activity of the K486N and H506D Lamin A/C IgF mutants (Supplementary Fig. 1d) and found that both showed substantially reduced nucleosome binding activity (Fig. 2b, e, lanes 9–10, c, d, f and g). We also tested the K486A and H506A Lamin A/C IgF mutants, in which the Lys486 and His506 residues are replaced by Ala. Like the patient-derived K486N mutant, the Lamin A/C IgF K486A mutant was defective in nucleosome binding (Fig. 2b, lane 11, and c). In contrast, unlike the patient-derived H506D mutant, the Lamin A/C IgF H506A mutant was only marginally defective in nucleosome binding (Fig. 2e, lane 11, and f). The acidic side chain of the Asp residue inserted at the Lamin A/C 506 position may repulse the DNA backbone phosphate at the nucleosomal dyad. In light of the fact that the K486N and H506D mutations of Lamin A/C are found in lipodystrophy patients, the defective nucleosome binding of the K486N and H506D mutants may be responsible for the chromatin disorganization phenotype.

The phosphorylation of the BAF Ser4 residue reportedly induces its dissociation from mitotic chromosomes, and the phosphomimetic BAF S4E mutant, in which the Ser4 residue is replaced by Glu, drastically reduces its DNA binding[27,33]. In the BAF-Lamin A/C-nucleosome complex, the BAF Ser4 residue is located close to the backbone of the linker DNAs (Fig. 1d). The BAF S4E mutant substantially decreased both the nucleosome binding and Lamin A/C IgF binding activities (Fig. 2h, lanes 6 and 8, i and j). Therefore, the linker DNA binding by BAF plays a pivotal role in the chromatin association of BAF and Lamin A/C.

### BAF-Lamin A/C connects two nucleosomes

We next conducted the cryo-EM analysis of the other BAF-Lamin A/C-nucleosome complex, with the GraFix fractions containing low mobility complexes (Fig. 1band Supplementary Fig. 1f). The analysis revealed that it contains two BAF-Lamin A/C-nucleosome complexes symmetrically connected by two additional BAF-Lamin A/C IgF molecules (Fig. 3aand Supplementary Fig. 2b). To bridge two nucleosomes containing BAF and Lamin A/C IgF, additional BAF dimers symmetrically bind the DNA at the superhelical location (SHL) ± 7.5 positions (about 75 base pairs from the dyad) of one nucleosome and at the

SHL ± 3 positions (about 30 base pairs from the dyad) of the other nucleosome (Fig. 3b). The orientation of the linker and nucleosomal DNAs bound to an additional BAF dimer is nearly vertical, and similar to that of the linker DNAs bound to the BAF dimer in the high gel mobility complex (Supplementary Fig. 3b). The positions of SHL ± 7.5 and SHL ± 3 allow the two BAF dimers to bind symmetrically to the nucleosomal DNAs with the perpendicular alignment around the SHL ± 7.5 and SHL ± 3 regions. The Lamin A/C IgF domain bound to the nucleosome-bridging BAF dimer does not contact the nucleosomal DNA, in contrast to the one bound to the nucleosomal dyad DNA (Fig. 3c). Therefore, the BAF dimer may have two distinct activities: securing intra-nucleosomal linker DNAs and compacting chromatin through inter-nucleosome bridging.

### BAF-Lamin A/C IgF connects chromatosomes containing the linker histone H1

Linker histone H1, the most abundant nucleosome-binding protein, forms tripartite binding interactions with the dyad and linker DNAs and compacts the nucleosome structure[34,35]. Since BAF-Lamin A/C IgF also binds to the nucleosomal dyad and linker DNAs, H1 may compete with BAF-Lamin A/C IgF for nucleosome binding. H1 not only ubiquitously exists within the nucleus, but also accumulates around the inner nuclear periphery, where BAF also resides in the interphase nuclei[36]. To elucidate how BAF binds to the nucleosome in the presence of H1, we performed the cryo-EM analysis of the H1-BAF-Lamin A/C IgF-nucleosome complex. The structure revealed that H1 binds the dyad and linker DNAs, forming a "chromatosome", and consequently eliminates the BAF-Lamin A/C IgF complex from the nucleosomal dyad and linker DNA regions (Fig. 3d, eand Supplementary Fig. 4). In this study, we used H1.1, which reportedly co-exists with BAF, for the cryo-EM analysis. The overall structure of the chromatosome in the BAF-Lamin A/C IgF-H1.1-nucleosome complex is quite similar to those of the chromatosomes containing H1.0, H1.4, or H1.10, although a slight difference exists (Supplementary Fig. 4d). Therefore, the H1 subtype may not affect the BAF-Lamin A/C binding to the chromatosome.

On the other hand, two BAF dimers complexed with the Lamin A/C IgF domain symmetrically bind to the DNA regions of the chromatosome at the SHL ± 7.5 and SHL ± 3 positions, and connect two chromatosomes (Fig. 3d, e). Therefore, BAF is capable of compacting chromatin together with H1. In the structure of the BAF-Lamin A/C IgF-H1-nucleosome complex, the distance between the nearest main chain regions of Lamin A/C IgF and H1 is approximately 8 Å, indicating that the globular domain of H1 and the IgF domain of Lamin A/C may not directly interact (Supplementary Fig. 4f).

## Discussion

BAF plays important roles in nuclear envelope reassembly and chromatin anchoring to the nuclear lamina during the cell cycle (Fig. 4). BAF reportedly associates with the core region of mitotic chromosomes in anaphase and functions in priming nuclear envelope reassembly (Fig. 4, left)[13,15]. At this stage, BAF may recruit components of the nuclear envelope and lamina, including Lamin A/C. Subsequently, the BAF molecules encircle the gathered chromosomes in telophase (Fig. 4, center). Finally, in interphase cells, the nuclear envelope and lamina are reassembled, and the chromosomes are anchored to the nuclear envelope and lamina (Fig. 4, right). In the present study, we determined three distinct structures of the BAF-Lamin A/C IgF-nucleosome complexes, which are all capable of tethering BAF and Lamin A/C to chromosomes.

In the first structure, the BAF-Lamin A/C IgF complex forms tripartite DNA binding interactions with the nucleosomal dyad and linker DNAs (Fig. 1c–e). We found a previously unrecognized DNA-binding interface of Lamin A/C, including the Lys486 and His506 residues (Fig.1d, e). Mutations of Lys486 or His506 in Lamin A/C have been detected in lipodystrophy patients who have disordered chromatin

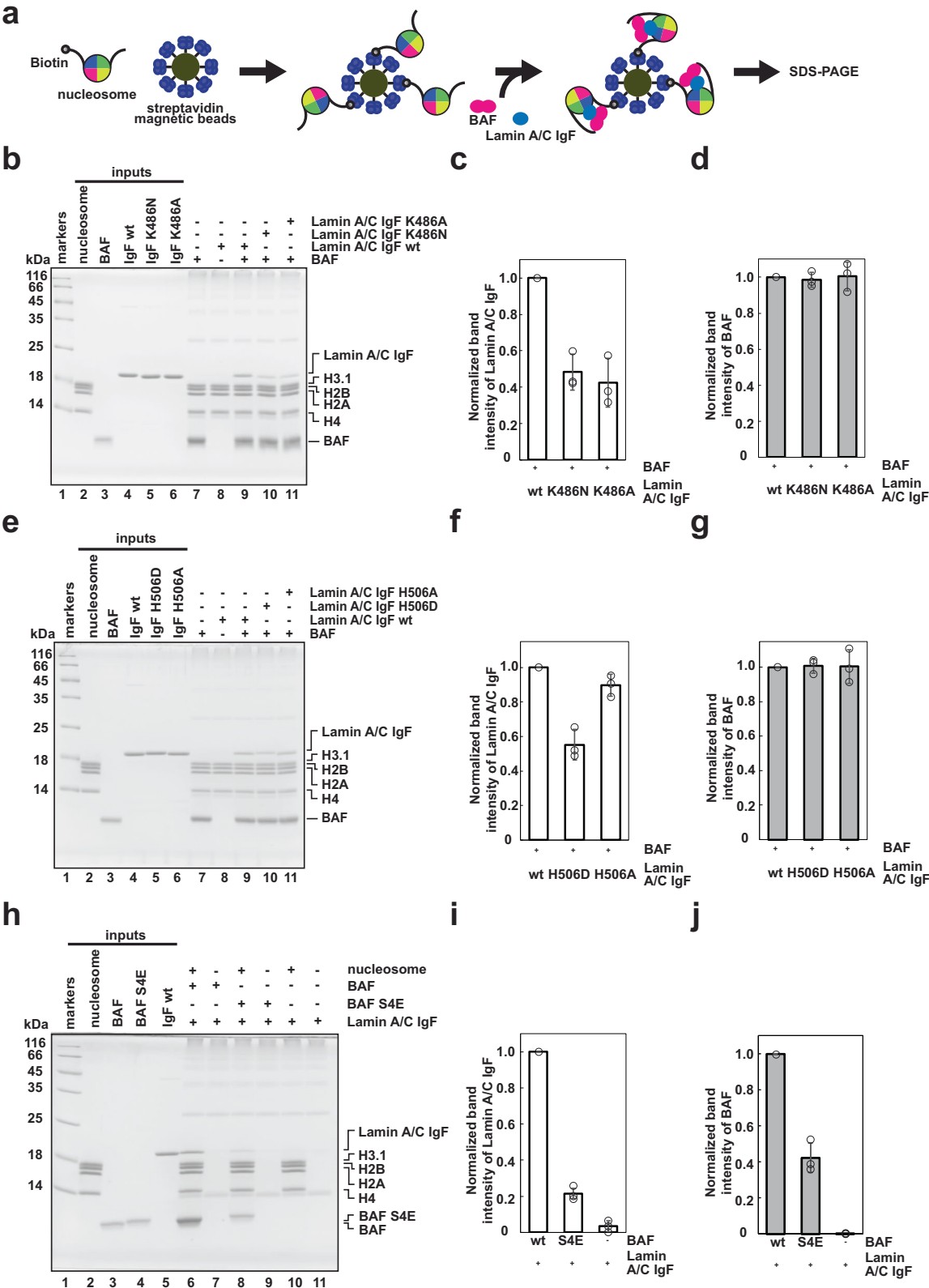

organization probably as a consequence of defective chromatin interactions with the nuclear lamina[29–31]. We determined that both the K486N and H506D mutations found in lipodystrophy patients decrease the nucleosome binding of Lamin A/C in the presence of BAF (Fig. 2). Therefore, the nucleosome binding by the Lys486 and His506 residues of Lamin A/C is functionally relevant, and the impaired nucleosome binding of the mutant Lamin A/C mutations may cause

defective chromatin anchoring to the nuclear lamina, as found in lipodystrophy patients' cells[32].

The higher-order chromatin architecture at the nuclear periphery is crucial for chromatin organization, regulating gene expression, and genome maintenance[37,38]. The nuclear lamina functions to anchor chromatin at the nuclear periphery[8,9]. BAF and Lamin A/C, which are co-localized at the nuclear periphery[16,18], collaboratively anchor

**Fig. 2 | Defective nucleosome binding of BAF and Lamin A/C IgF mutants.**
**a** Schematic representation of the pull-down assay with the biotinylated nucleosome. **b** BAF and Lamin A/C IgF (or Lamin A/C IgF K486N or K486A mutant) were incubated with the biotinylated nucleosome and captured with streptavidin-magnetic beads. Proteins eluted from the streptavidin-magnetic beads were analyzed by SDS-PAGE with Coomassie Brilliant Blue (CBB) staining. **c**, **d** Quantification of the nucleosome binding by BAF and Lamin A/C IgF in panel (**b**). The band intensities of Lamin A/C IgF mutants (**c**) and BAF (**d**) (lanes 10 and 11) were normalized to those in lane 9. Bar graphs and circles show the means and individual values of three independent experiments, respectively. Error bars represent the standard deviation. **e** BAF and Lamin A/C IgF (or Lamin A/C IgF H506D or H506A mutant) were incubated with the biotinylated nucleosome and captured with streptavidin-magnetic beads. Proteins eluted from the streptavidin-magnetic beads

were analyzed by SDS-PAGE with CBB staining. **f**, **g** Quantification of the nucleosome binding by BAF and Lamin A/C IgF in panel (**d**). The band intensities of Lamin A/C IgF mutants (**f**) and BAF (**g**) (lanes 10 and 11) were normalized to those in lane 9. Bar graphs and circles show the means and individual values of three independent experiments, respectively. Error bars represent the standard deviation. **h** BAF (or BAF S4E) and Lamin A/C IgF were incubated with the biotinylated nucleosome and captured with streptavidin-magnetic beads. Proteins eluted from the streptavidin-magnetic beads were analyzed by SDS-PAGE with CBB staining. **i**, **j** Quantification of the nucleosome binding by BAF and Lamin A/C IgF in panel (**h**). The band intensities of Lamin A/C IgF (**i**) and the BAF mutant (**j**) (lanes 8 and 10) were normalized to those in lane 6. Bar graphs and circles show the means and individual values of three independent experiments, respectively. Error bars represent the standard deviation. Source data are provided as a Source Data file.

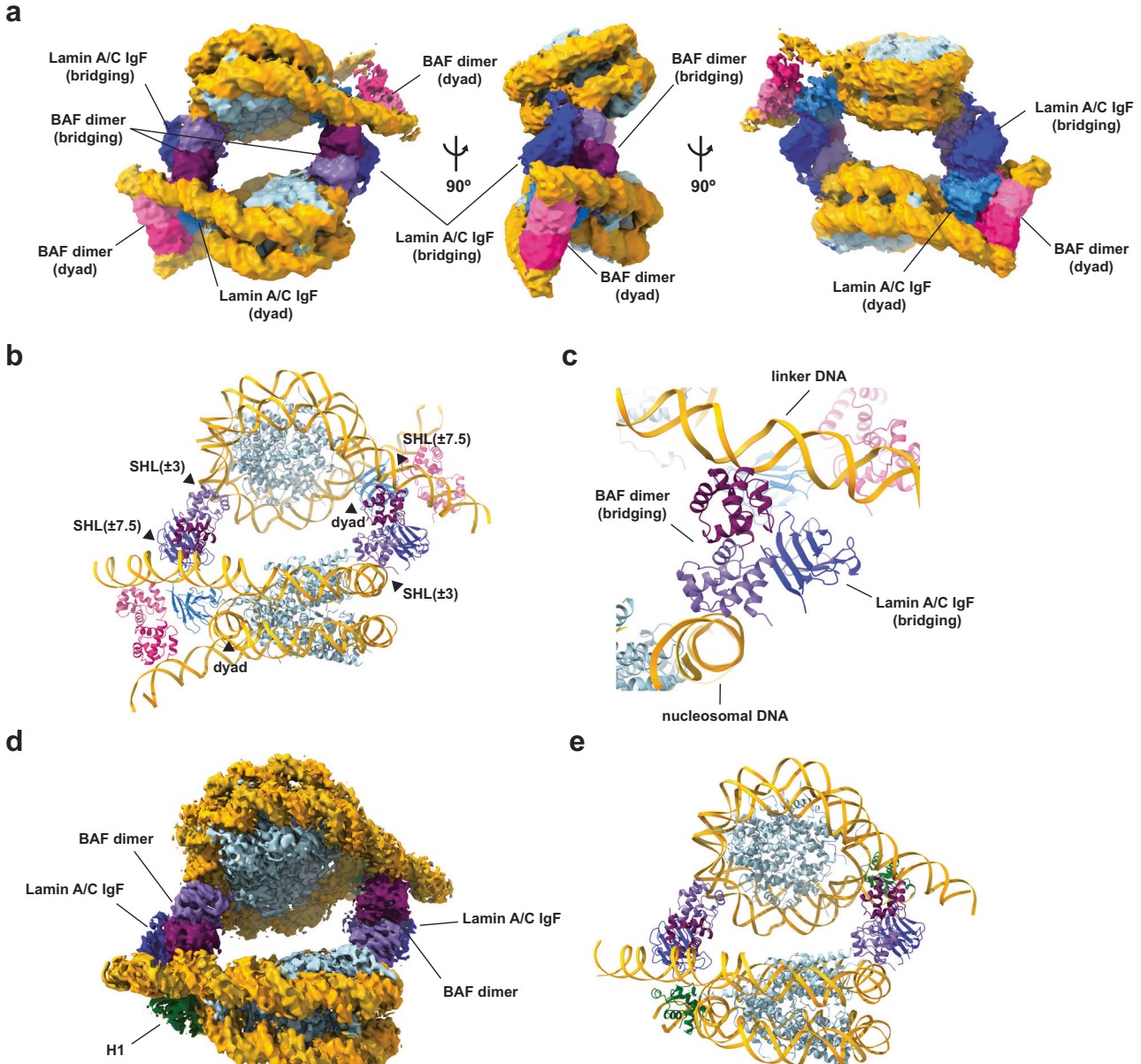

**Fig. 3 | Cryo-EM structures of nucleosomes and chromatosomes connected by the BAF-Lamin A/C IgF complexes. a** Cryo-EM map of the BAF-Lamin A/C IgF-nucleosome complex of fraction 2 in Fig. 1 (panel **b**). Histones and DNA are colored light blue and gold, respectively. BAFs and Lamin A/C IgFs for the dyad binding and the nucleosome bridging are indicated in the cryo-EM maps. **b** Structure of the BAF-Lamin A/C IgF-nucleosome complex in panel (**a**). **c** Close-

up view of the interface between two nucleosomes and the BAF dimer and Lamin A/C IgF. **d** Cryo-EM map of the BAF-Lamin A/C IgF-H1-nucleosome complex. Histones and DNA are colored light blue and gold, respectively. H1 bound to the nucleosomal dyad, and BAFs and Lamin A/C IgFs involved in the nucleosome bridging are indicated in the cryo-EM maps. **e** Structure of the BAF-Lamin A/C IgF-H1-nucleosome complex in panel (**d**).

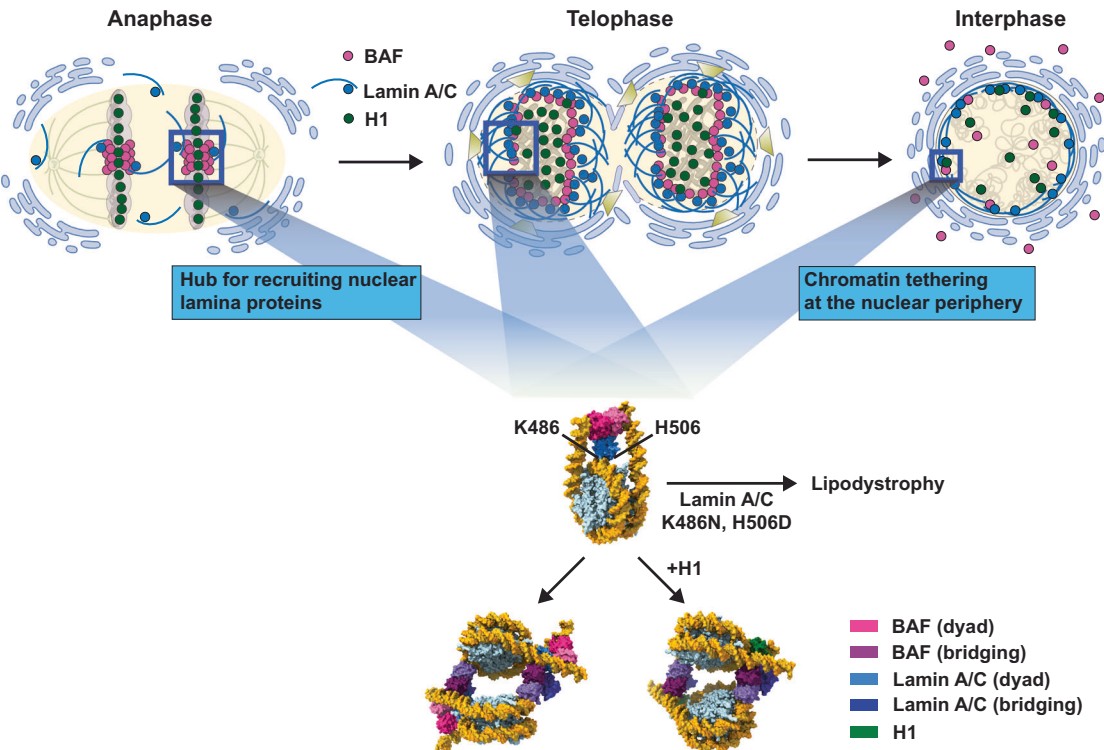

**Fig. 4 | Schematic models for BAF and Lamin A/C functions through the cell cycle.** In anaphase (top left panel), BAF accumulates at the core region of mitotic chromosomes and recruits nuclear lamina proteins for reassembly of the nuclear lamina and envelope. In telophase (top center panel), BAF encircles chromosomes with other nuclear lamina proteins. In the interphase (top right panel), a fraction of BAF associated with Lamin A/C is located at the nuclear periphery. The middle center panel shows the structure of the BAF-Lamin A/C IgF-nucleosome complex, in which BAF and the Lamin A/C IgF domain bind to the nucleosomal dyad. The K486 and H506 residues of Lamin A/C directly interact with the DNA at the nucleosomal dyad. Since the K486N and H506D mutants of Lamin A/C potentially cause lipodystrophy by their impaired nucleosome binding, this structure may function in chromatin association in anaphase, telophase, and interphase during the cell cycle. BAF also has the ability to bridge nucleosomes. This nucleosome bridging activity of BAF may function to form a specific chromatin architecture. The linker histone H1 competes with BAF-Lamin A/C at the nucleosomal dyad and induces nucleosome bridging by BAF. Created in BioRender. Horikoshi, N. (2025) https://BioRender.com/u03n525.

chromatin to the nuclear lamina, and function in chromatin organization, nuclear envelope assembly, and nuclear envelope repair[18,39]. There are chromatin regions near the nuclear periphery with and without linker histones[40]. In those without linker histones, BAF-Lamin A/C may bind to the nucleosomal linker and dyad DNA regions, and thereby function to capture chromatin. Meanwhile, in the chromatin regions with linker histones, H1 or another linker histone probably competes with BAF-Lamin A/C bound to the nucleosomal dyad, and may promote the BAF-mediated nucleosome bridging (Fig. 3). Therefore, in both chromatin regions without or with linker histones, the BAF dimer binds chromatin by the "tripartite nucleosome-binding" mode or the "nucleosome bridging" mode, and both could capture the IgF domain of Lamin A/C to anchor chromatin to the nuclear periphery. In the inner nuclear periphery, BAF reportedly binds to LEM domain proteins such as emerin[13,16]. Superimposition of the previous BAF-Lamin A/C IgF-emerin LEM domain complex on the present BAF-Lamin A/C IgF-nucleosome complexes revealed that emerin does not cause a steric clash (Supplementary Fig. 5). These findings suggest that the present BAF-Lamin A/C IgF-nucleosome complexes would also allow emerin binding without structural rearrangement. Further studies are awaited to understand the distinct functions of these three different types of BAF-Lamin A/C IgF-nucleosome complexes, which may have specific and/or common roles in a cell-cycle-dependent manner.

The nuclear lamina proteins, including BAF, are closely associated with progeria syndrome and muscular dystrophy, which are collectively referred to as laminopathies[41]. The cryo-EM structures of the BAF-Lamin A/C IgF-nucleosome complexes, with and without the linker histone H1, would pave the way toward understanding the disease mechanisms and contribute to the development of therapeutic strategies for laminopathies.

## Methods

### Purification of Barrier-to-autointegration factor (BAF)
Human BAF and the BAF S4E mutant were purified by the previously reported method, with minor modifications[16]. Briefly, the *Escherichia coli* (*E. coli*) Rosetta 2 (DE3) strain (Novagen) expressing the N-terminally hexahistidine-tag fused human BAF (His₆-BAF) was lysed by sonication in 50 mM Tris-HCl (pH 8.0) buffer, containing 0.5 M sodium chloride, 5% glycerol, and 1 mM PMSF. After centrifugation, the pellet was denatured with 50 mM Tris-HCl (pH 8.0) buffer, containing 0.5 M sodium chloride, 5% glycerol, and 7 M urea. His₆-BAF was purified from the supernatant by Ni-NTA agarose resin (Qiagen). For refolding, purified His-BAF was dialyzed against Tris-HCl (pH 7.5) buffer, containing 0.2 M sodium chloride, 1 mM EDTA, and 2 mM 2-mercaptoethanol. The hexahistidine-tag was cleaved by thrombin protease, and then BAF was further purified by HiLoad 16/600 Superdex 75 chromatography in 20 mM HEPES-NaOH (pH 7.5) buffer containing 150 mM sodium chloride and 1 mM dithiothreitol. The purified BAF was flash-frozen with liquid nitrogen and stored at − 80 °C.

### Purification of the Ig-fold domain of Lamin A/C
Human Lamin A/C (411–566) was purified by the previously reported method, with minor modifications[16]. Briefly, the *E. coli* Rosetta 2 (DE3) strain expressing the N-terminally MBP-tag fused Lamin A/C

(411–566) (MBP-Lamin A/C IgF) was lysed by sonication in 50 mM Tris-HCl (pH 8.0) buffer, containing 0.5 M sodium chloride, 5% glycerol, 1 mM PMSF, 1 mM EDTA, and 2 mM 2-mercaptoethanol. MBP-Lamin A/C IgF was purified using amylose resin (New England Biolabs). After MBP-tag cleavage on-column with TEV protease, Lamin A/C IgF was further purified by HiLoad 16/600 Superdex 75 chromatography in 20 mM HEPES-NaOH (pH 7.5) buffer, containing 150 mM sodium chloride and 1 mM dithiothreitol. The purified Lamin A/C IgF was flash-frozen with liquid nitrogen and stored at − 80 °C.

## Purification of linker histone H1

Human linker histone H1.1 was purified according to the previously reported method[42,43]. Briefly, the $His_6$-SUMO-tag fused H1.1 was bacterially expressed and purified with Ni-NTA agarose resin. After $His_6$-SUMO-tag cleavage, H1.1 was further purified by Mono S (Cytiva) column chromatography. H1.1 was finally dialyzed against 20 mM Tris-HCl (pH 7.5), 100 mM sodium chloride, 2 mM 2-mercaptoethanol, and 10% glycerol. The purified H1.1 was flash-frozen with liquid nitrogen and stored at − 80 °C.

## Preparation of nucleosome

Histones H2A, H2B, H3, and H4 were bacterially expressed and purified as described previously[44]. Nucleosomes were reconstituted with the purified histone octamer and a 193 base-pair DNA fragment (5′-ATCG-GACCCTATCGCGAGCCAGGCCTGAGAATCCGGTGCCGAGGCCGCTCAATTGGTCGTAGACAGCTCTAGCACCGCTTAAACGCACGTACGCGCTGTCCCCCGCGTTTTAACCGCCAAGGGGATTACTCCCTAGTCTCCAGGCACGTGTCAGATATATACATCCAGGCCTTGTGTCGCGAAATTCATAGAT-3′), containing the Widom 601 sequence, by the salt dialysis method. For the biotinylated nucleosome, an oligo DNA primer (5′-ATCGGACCC-TATCGCGAGCCAG-3′) biotinylated at the 5′-end and an unmodified oligo DNA primer (5′-ATCTATGAATTTCGCGACACAAGGCCT-3′) (FASMAC) were used for the synthesis of the abovementioned 193 base-pair of DNA fragment. The synthesized DNA was extracted with phenol-chloroform-isoamyl alcohol, and then further purified using a Model 491 Prep Cell apparatus (Bio-Rad). Reconstituted nucleosomes were purified using the Prep Cell apparatus. Purified nucleosomes were dialyzed against 20 mM Tris-HCl (pH 7.5), containing 1 mM dithiothreitol and 5% glycerol, flash-frozen with liquid nitrogen, and then stored at − 80 °C.

## Nucleosome pull-down assay

Streptavidin magnetic beads (10 μl, Sigma-Aldrich) were pre-washed three times with 500 μl of W1 buffer, containing 20 mM HEPES-NaOH (pH 7.5), 200 mM sodium chloride, 0.2 mM EDTA, 5% glycerol, 0.1% NP-40, and 1 mM dithiothreitol. The nucleosome containing the biotinylated 193 base-pair DNA (10 μg) was conjugated to the pre-washed streptavidin magnetic beads in a 200 μl reaction mixture, containing 20 mM HEPES-NaOH (pH 7.5), 75 mM sodium chloride, 0.2 mM EDTA, 5% glycerol, 0.1% NP-40, and 0.1 mM dithiothreitol, by gentle rotation for 1 h at 4 °C. The nucleosome-conjugated magnetic beads were washed three times with 500 μl of W2 buffer, containing 20 mM HEPES-NaOH (pH 7.5), 75 mM sodium chloride, 0.2 mM EDTA, 5% glycerol, 0.1% NP-40, and 1 mM dithiothreitol. BAF or the BAF S4E mutant at a 4-fold molar ratio and Lamin A/C IgF wild-type or the K486N/A, H506D/A mutants at an 8-fold molar ratio to the nucleosomes were then mixed with the nucleosome-conjugated magnetic beads and incubated for 2 h at 4 °C. Afterwards, the beads were washed with W2 buffer three times. Bound proteins were eluted by mixing with 40 μl of 1 × SDS sample buffer, containing 62.5 mM Tris-HCl (pH 8.0), 2% SDS, 10% glycerol, 0.005% (w/v) bromophenol blue, and 0.1 M 2-mercaptoethanol, and incubating at 95 °C for 2 min. The samples were analyzed by 18% SDS-polyacrylamide gel electrophoresis with CBB staining. The uncropped data are shown in the Source Data file.

## Sample preparation for cryo-EM

Purified frozen nucleosomes, BAF, and Lamin A/C IgF, with or without H1.1, were thawed and incubated together for 30 min at 30 °C. For BAF-Lamin A/C IgF-nucleosome complexes, BAF at a 3-fold molar ratio and Lamin A/C IgF at a 6-fold molar ratio were mixed with 0.93 μM nucleosomes in a 422 μl reaction mixture, containing 14 mM Tris-HCl (pH 7.5), 7.7 mM HEPES-NaOH (pH 7.5), 39 mM sodium chloride, 0.95% glycerol, and 1 mM dithiothreitol. For BAF-Lamin A/C IgF-H1-nucleosome complexes, BAF at a 3-fold molar ratio, Lamin A/C IgF at a 6-fold molar ratio, and H1.1 at a 2-fold molar ratio were mixed with 0.93 μM nucleosomes in a 422 μl reaction mixture, containing 15 mM Tris-HCl (pH 7.5), 7.8 mM HEPES-NaOH (pH 7.5), 46 mM sodium chloride, 1.3% glycerol, 0.93 mM dithiothreitol, and 0.15 mM 2-mercaptoethanol. The resulting mixtures were loaded on a 5-20% continuous sucrose gradient with 0-0.1% glutaraldehyde[45]. The samples were centrifuged at 125,000 × g for 16 h at 4 °C, using an SW41Ti rotor (Beckman Coulter). Fractions were taken from the top of the solution and analyzed by native-polyacrylamide gel electrophoresis with ethidium bromide staining. Fractions containing nucleosomes, BAF, and Lamin A/C IgF, with or without H1.1, were collected. The samples were desalted on a PD-10 column (Cytiva) with 20 mM HEPES-NaOH (pH 7.5) buffer, containing 50 mM sodium chloride and 1 mM dithiothreitol. The samples were concentrated with an Amicon Ultra Centrifugal Filter (MWCO: 30 kDa). The concentrations of BAF-Lamin A/C IgF-nucleosome (high gel mobility), BAF-Lamin A/C IgF-nucleosome (low gel mobility), and BAF-Lamin A/C IgF-H1-nucleosome, calculated based on the absorbance at 260 nm and converted to dsDNA, were 267.8 ng/μl, 164.3 ng/μl, and 303.4 ng/μl, respectively.

## Grid preparation for cryo-EM

For the BAF-Lamin A/C IgF-nucleosome complex and the BAF-Lamin A/C IgF-H1.1-nucleosome complex, 2.5 μl portions of the samples were loaded on glow-discharged Quantifoil R1.2/1.3 Au or Cu, 200-mesh grids. The grids were blotted with blot force 0 for 4–8 sec at 4 °C in 100% humidity, and then plunge-frozen in liquid ethane by using a Vitrobot Mark IV (Thermo Fisher Scientific).

## Cryo-EM data collection

Cryo-EM data were collected with a Krios G4 Cryo-TEM (Thermo Fisher Scientific) operating at 300 kV and a magnification of 81,000 ×, equipped with a BioQuantum K3 direct electron detector (Gatan) and an energy filter. The defocus range was from − 1.25 to − 2.5 μm. Data collection conditions are also shown in Supplementary Table 1.

## Image processing

Image processing was performed using the Relion4 software[46]. Frames of the movies were aligned by MOTIONCOR2 with dose weighting[47]. The contrast transfer function was estimated with CTFFIND4[48]. For the BAF-Lamin A/C IgF-nucleosome complex (Fraction 1), 10,189 micrographs were collected, and 5,720,577 particles were picked by Laplacian-of-Gaussian (LOG)-based auto-picking. Picked particles were extracted from micrographs with 4 × binning. The 2D classification was performed to remove junk particles. The initial model was generated de novo, and then two rounds of 3D classification were performed. After removing the binning, CTF refinement and Bayesian polishing were performed. The final map had a global resolution of 3.82 Å, which was estimated from the gold standard Fourier Shell Correlation (FSC = 0.143) criteria[49]. For the BAF-Lamin A/C IgF-nucleosome complex (Fraction 2), 5779 micrographs were collected, and 2,697,637 particles were picked by template-based auto-picking, using the 3D class averages of auto-picked particles based on a LOG filter as the template. Picked particles were extracted from micrographs with 4 × binning. The 2D classification was performed to remove junk particles. After one round of 3D classification, the binning was removed, and two

more rounds of 3D classification were performed. The final map had a global resolution of 7.14 Å, which was estimated from the gold standard Fourier Shell Correlation (FSC = 0.143) criteria.

For the BAF-Lamin A/C IgF-H1-nucleosome complex, 5486 micrographs were collected, and 3,105,378 particles were picked by template-based auto-picking, using the 3D class averages of auto-picked particles based on a LOG filter as the template. Picked particles were extracted from micrographs with 4 × binning. The 2D classification was performed to remove junk particles. After two rounds of 3D classification, the binning was removed. After one more round of 3D classification, CTF refinement, and Bayesian polishing were performed. The final map had a global resolution of 4.50 Å, which was estimated from the gold standard Fourier Shell Correlation (FSC = 0.143) criteria. A mask was created around the chromatosome, and 3D refinement and CTF refinement were performed. The final map for the chromatosome had a global resolution of 3.57 Å, which was estimated from the gold standard Fourier Shell Correlation (FSC = 0.143) criteria. The local resolution was estimated by Relion4. Detailed processes are shown in Supplementary Figs. 6–8.

### Model building and refinement
Model building was performed with COOT[50], using the initial model structures of the nucleosome (7K5X[35] and 5B0Z[51]), BAF (6GHD[16]), Lamin A/C IgF (6GHD[16]), and H1.1 (AlphaFold2[52–54]). The atomic coordinates of each complex were refined with ISOLDE[55] and Phenix[56]. Structural validation was performed with MolProbity[57]. Statistics for refinement and validation of models are shown in Supplementary Table 1.

### Figures
Figures were created with Pymol, ChimeraX, BioRender.com, and Adobe Illustrator 2024.

### Reporting summary
Further information on research design is available in the Nature Portfolio Reporting Summary linked to this article.

## Data availability
The cryo-EM maps and atomic coordinates of the BAF-Lamin A/C IgF-nucleosome (Fractions 1 and 2) and BAF-Lamin A/C IgF-H1-nucleosome complexes were deposited in the Electron Microscopy Data Bank (EMDB) and the Protein Data Bank (PDB), respectively. Accession codes of these cryo-EM maps are EMD-61231 (BAF-Lamin A/C IgF bound to the nucleosome (High mobility complex)), EMD-61232 (BAF-Lamin A/C IgF bound to the nucleosome (Low mobility complex)) and EMD-61233 (BAF-Lamin A/C IgF bound to the chromatosome containing H1.1). Accession codes of these atomic coordinates are 9J8M (BAF-Lamin A/C IgF bound to the nucleosome (High mobility complex)), 9J8N (BAF-Lamin A/C IgF bound to the nucleosome (Low mobility complex)) and 9J8O (BAF-Lamin A/C IgF bound to the chromatosome containing H1.1). Uncropped images are shown in the Source data file. Source data are provided in this paper.

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

## Acknowledgements

We thank Y. Iikura, Y. Takeda, M. Dacher, and L. Negishi (Univ. Tokyo) for their assistance. This work was supported by the Japan Society for the Promotion of Science (KAKENHI grants JP22K06076, JP23K24013, and JP24H02328 to N.H.; JP22K06098 to Y.T.; JP23H05475, JP24H02319, and JP24H02328 to H.K.); Japan Science and Technology Agency ERATO (grant JPMJER1901 to H.K.); and the Platform Project for the Research Support Project for Life Science and Drug Discovery (BINDS) from AMED (grant JP24ama121009 to H.K.; JP24ama121003 to Y.T.); and Japan Science and Technology Agency, CREST Grant Number JPMJCR24T3 (to H.K.).

## Author contributions

N.H., R.M., and C.S.-F. conducted sample preparation and biochemical analysis. N.H., R.M., M.O., and Y.T. collected cryo-EM data and determined structures. N.H. and H.K. conceived, designed, and supervised this study. N.H. prepared all figures. N.H., Y.T., and H.K. wrote the manuscript. All authors discussed and commented on the manuscript.

## Competing interests

The authors declare no competing interests.
