## [Peer Review file · Nature Communications]

Cryo-EM structures of the BAF-Lamin A/C complex bound to nucleosomes

Corresponding Author: Professor Hitoshi Kurumizaka

Version 0:

Reviewer comments:

Reviewer #1

(Remarks to the Author)

The manuscript entitled "Cryo-EM structures of the BAF-Lamin A/C complex bound to nucleosome" by Horikoshi and colleagues investigates the structural basis of the BAF-Lamin A/C complex binding to mono-nucleosomes with linker DNA. Lamins form a bridge between the inner nuclear envelope and chromatin, a crucial role evidenced by the numerous mutations causing laminopathies. Previous studies have shown that the Lamin A/C Ig domain directly binds the dimeric chromatin-associated BAF on one side and emerin, a nuclear envelope protein, on the other side (Samson et al. NAR 2018; Cai et al. JBC 2007). The dimeric BAF protein uniquely binds and bridges DNA in a sequence-independent manner (Lee et al. PNAS 1998; Bradley et al. NSMB 2005).

In this work, the authors use cryo-EM to analyze how BAF and the Lamin A/C IgF domain bind to nucleosomes. Their initial structure reveals that the BAF/Lamin complex exhibits two binding modes to the nucleosome: it bridges two mono-nucleosomes and binds at the nucleosomal dyad DNA, where the linker histone also binds. Therefore, the authors have determined the cryo-EM structure of the BAF/Lamin complex bound to mono-nucleosomes, which have the linker histone H1 bound at the dyad. This structure demonstrates that two BAF/Lamin complexes can still bridge two nucleosomes. It is exciting to see that nucleosomes are not inhibitory to BAF binding, and that BAF does not rely solely on linker DNA regions to bridge chromatin. Furthermore, the authors use an elegant nucleosome pull-down assay to study the impact of patient-derived mutations in the Lamin A/C IgF domain on nucleosome binding, as well as how PTM of BAF influence nucleosome binding.

Overall, the presented data are of very high quality. It is a well-written and easy-to-follow study, but the authors could communicate more clearly what new insights are gained from this study compared to previously published structures (Samson et al. NAR 2018; Bradley et al. NSMB 2005).

The authors have high-quality cryo-EM data and should demonstrate how BAF and lamin proteins, as well as histones and DNA, fit into the experimental densities in the supplemental data. Currently, the fit of histone H1 into the cryo-EM map is shown (Supplementary Figure 2b).

Previous structural work has shown that dimeric BAF bridges two DNAs in a sequence-independent manner by binding to the minor groove and orienting the two DNA molecules approximately perpendicular to each other. How different is BAF binding to nucleosome linker DNA from its binding to free DNA? Similarly, is BAF binding to nucleosomal DNA (SHL ± 3 and SHL ± 7.5) during the bridging of two nucleosomes related to its binding to free DNA? How do these structures compare? A simple overlay and comparison of these structures should be included in the manuscript.

In both structures of the bridged nucleosomes, the BAF/Lamin complex binds to the same SHL locations. Are there structural constraints that would prevent BAF from binding to other locations, such as histone tails? Additionally, in both structures, BAF binds one DNA strand within the nucleosome particle (SHL ± 3), while the other location is in the linker DNA. Is it structurally necessary for BAF to have at least one DNA in the linker region, not bound by histone proteins? Can BAF bridge two nucleosomes without any linker DNA?

It would be interesting to know if the inner nuclear membrane protein emerin could bind to the BAF-nucleosome complex, given its role in tethering chromatin to the nuclear periphery/nuclear envelope. It should be straightforward for the authors to overlay their structure of the BAF-Lamin A/C complex bound to the nucleosome with the previously published structure of the ternary complex involving Lamin A/C, BAF, and emerin. Are there any structural clashes that would prevent emerin from binding to BAF in the structures described in this manuscript? In other words, are structural rearrangements of BAF necessary while bound to the nucleosome in order to bind emerin, or can emerin bind to the structures as presented in the manuscript? It would be interesting to address this point in the discussion.

The authors should improve the methods section: as it stands, it would be impossible for another scientist to reproduce their

work. For example, it is unclear if the proteins can be frozen or not, and the same applies to the cryo-EM sample. The methods section lacks concentration details for any sample in all methods. For instance, in the nucleosome pull-down assay, there is no information on how much bead material was used, how much sample, their concentrations and volumes, or the ratio of nucleosomes to binding proteins; the same applies to other methods sections.

Minor comments:

1. The authors sometimes write Lamin A/C Lys483, and other times Lamin A/C Lys486.
2. In Supplementary Figure 5c, the number of particles for the initial 3D classification is missing.
3. What is the unmapped gray density in Figure 3a (middle and right panels) between the DNA and the dark blue Lamin A/C?
4. The gel showing the quality of the BAF-Lamin A/C IgF-H1-nucleosome complex should be shown.

Reviewer #2

(Remarks to the Author)

The association between mitotic chromosomes and the nuclear envelope is a critical aspect of cell biology, yet its structural details remain largely unexplored. Previous studies have identified that Barrier-to-autointegration factor (BAF) anchors chromatin to the nuclear lamina via Lamin A/C. In this manuscript, the authors present a structural analysis of BAF and Lamin A/C binding to the nucleosome using cryo-EM. They discovered that the Lamin A/C IgF domain binds to the nucleosome dyad, while BAF interacts with both the Lamin A/C IgF domain and linker DNA. Additionally, they demonstrated that histone H1 can substitute for Lamin A/C. Notably, the study reveals that the BAF-Lamin A/C complex can bridge two nucleosomes or chromatosomes. Furthermore, mutations in the Lamin A/C residues Lys483 and His506, which are associated with lipodystrophy, were found to significantly impair nucleosome binding.

The structures elucidated in this study provide valuable insights into how BAF, Lamin A/C, and histone H1 associate with nucleosomes, which may have broader implications for chromatin organization within the nucleus. The findings are both novel and functionally significant.

Technical Comments:

1. Cross-Linking Concerns: A primary concern is the use of cross-linking in the cryo-EM study, which could potentially introduce artifacts. To address this, the authors should demonstrate that the nucleosome or chromatosome dimers exist in solution independently of cross-linking. Experiments such as gel filtration, analytical ultracentrifugation, small-angle light scattering, or generating a low-resolution cryo-EM density map without cross-linking would be beneficial.
2. Lamin A/C Full-Length Protein: The study utilized only the Lamin A/C IgF domain, not the full-length protein. Conducting binding assays with the full-length Lamin A/C, if it can be expressed and purified, would enhance the impact of the paper by providing a more comprehensive understanding.
3. Binding at SHL ± 3 and SHL ± 7.5 : The manuscript should provide an explanation for the observed binding at SHL ± 3 and SHL ± 7.5 positions.
4. Clarification in Extended Data Figure 2: It is unclear whether Lamin A/C IgF interacts with H1 in Extended Data Figure 2. Clarification of this interaction is needed.
5. Comparison with Chromatosome Structures: A comparison of the H1.1 chromatosome structure reported here with previously published high-resolution structures of chromatosomes containing either H1.0 or H1.4 (Zhou et al., Mol Cell, 2021; PMID: 33238161) would be useful. This could help determine whether the Lamin A/C IgF domain affects H1 binding during chromatosome dimer formation.

Reviewer #3

(Remarks to the Author)

Barrier-to-autointegration factor (BAF) binds to mitotic chromosomes and upon mitosis exit assists the recruitment of lamins and other proteins for the assembly of both the lamina and the nuclear envelope and in anchoring chromatin to the nuclear envelope via the lamina. This is realized through specific interactions with lamin A/C. Yet, the underlying mechanism of this process remained obscure. The manuscript aims to solve the structure of the BAF-Lamin A/C complex bound to nucleosomes and thus, to shed light on the lamina assembly and chromatin anchorage. The authors have initially solved the Cryo-EM structure of the BAF-Lamin A/C IgF-nucleosome complex by using highly purified recombinant proteins and reconstituted nucleosomes. The data showed that BAF-Lamin A/C binds the nucleosome by tripartite DNA binding and that the nucleosome binding of the lamin A/C IgF domain is dependant of the binding of the BAF dimer. This was followed by mutational analysis of nucleosome binding by BAF and lamin A/C and subsequent determination of the nucleosome complexes assembled with the mutated proteins. The results evidence that the K486N and H506D mutations of lamin A/C led to defective nucleosome binding. Since these mutations are observed in lipodystrophy patients, this could be associated with the aetiology of the disease. In addition, the authors showed that BAF-Lamin A/C complexes were able to connect two nucleosomes.

This is a nice manuscript which addresses an important problem and successfully solves it. From technical point of view, the

manuscript is perfect. The experiments are well designed and professionally carried out. The text is easy to read and understand. I have only few minor notes.

1. Lanes 82-83: "...which may correspond to the high gel mobility complex...". This sentence should be rewritten. In order to claim this, the different complexes should be purified by glycerol/sucrose gradient and then separated on a native gel, which will identify the mobility of the complexes.

2. Lanes 84, 86 : It would be better to remove "interestingly" and "surprisingly" from these lanes as well as throughout the text.

3. Lane 87: ".Lys483 and His506.". Lys483 should be replaced with Lys486.

4. lane 122, 123: idem point 1.

5. lane 136: "...compacts the nucleosome structure 33,34. References 33&34 are not the most appropriate ones and should be substituted with references that reflect the claim "...forms tripartite binding interactions with the dyad and linker DNAs and compacts the nucleosome structure".

6. lane 189: "will pave" should be replaced with "would pave".

7. lane 377, 378: the length of the linker between the biotin and the base used for the preparation of the 193 bp Widom sequence should be mentioned.

8. lane 391, 392: "Nucleosomes, BAF, and Lamin A/C IgF, with or without H1.1, were mixed and incubated at 30°C for 30 min". The composition of the buffer used in the binding reaction has to be described. Are the nucleosomes containing already H1.1 or H1.1 is added together with BAF and Lamin A/C IgF? How the H1.1 bound nucleosomes are prepared?

Version 1:

Reviewer comments:

Reviewer #1

(Remarks to the Author)

The authors have answered to all raised questions and comments.
I fully support the publication of the manuscript in the current form.

Reviewer #2

(Remarks to the Author)

My concerns have been addressed in the revised manuscript. I recommend its publication.

REVIEWER COMMENTS

Reviewer #1 (Remarks to the Author):

The manuscript entitled “Cryo-EM structures of the BAF-Lamin A/C complex bound to nucleosome” by Horikoshi and colleagues investigates the structural basis of the BAF-Lamin A/C complex binding to mono-nucleosomes with linker DNA.

Lamins form a bridge between the inner nuclear envelope and chromatin, a crucial role evidenced by the numerous mutations causing laminopathies. Previous studies have shown that the Lamin A/C Ig domain directly binds the dimeric chromatin-associated BAF on one side and emerin, a nuclear envelope protein, on the other side (Samson et al. NAR 2018; Cai et al. JBC 2007). The dimeric BAF protein uniquely binds and bridges DNA in a sequence-independent manner (Lee et al. PNAS 1998; Bradley et al. NSMB 2005).

In this work, the authors use cryo-EM to analyze how BAF and the Lamin A/C IgF domain bind to nucleosomes. Their initial structure reveals that the BAF/Lamin complex exhibits two binding modes to the nucleosome: it bridges two mono-nucleosomes and binds at the nucleosomal dyad DNA, where the linker histone also binds. Therefore, the authors have determined the cryo-EM structure of the BAF/Lamin complex bound to mono-nucleosomes, which have the linker histone H1 bound at the dyad. This structure demonstrates that two BAF/Lamin complexes can still bridge two nucleosomes. It is exciting to see that nucleosomes are not inhibitory to BAF binding, and that BAF does not rely solely on linker DNA regions to bridge chromatin. Furthermore, the authors use an elegant nucleosome pull-down assay to study the impact of patient-derived mutations in the Lamin A/C IgF domain on nucleosome binding, as well as how PTM of BAF influence nucleosome binding.

Overall, the presented data are of very high quality. It is a well-written and easy-to-follow study, but the authors could communicate more clearly what new insights are gained from this study compared to previously published structures (Samson et al. NAR 2018; Bradley et al. NSMB 2005). The authors have high-quality cryo-EM data and should demonstrate how BAF and lamin proteins, as well as histones and DNA, fit into the experimental densities in the supplemental data. Currently, the fit of histone H1 into the cryo-EM map is shown (Supplementary Figure 2b).

Reply)

Thank you very much for this comment. We added panels to Supplementary Figure 2, in which models of BAF-lamin A/C IgF-nucleosome complexes are fit into the density maps.

Previous structural work has shown that dimeric BAF bridges two DNAs in a sequence-independent manner by binding to the minor groove and orienting the two DNA molecules approximately perpendicular to each other. How different is BAF binding to nucleosome linker DNA from its binding to free DNA? Similarly, is BAF binding to nucleosomal DNA (SHL ± 3 and SHL ± 7.5) during the bridging of two nucleosomes

related to its binding to free DNA? How do these structures compare? A simple overlay and comparison of these structures should be included in the manuscript.

Reply)

Thank you very much for these insightful comments. In the revised manuscript, we included figures that compare the orientations of the DNAs in the low gel mobility complex, the high gel mobility complex, and the BAF-DNA complex, in the new Supplementary Figure 3.

In the BAF-Lamin A/C IgF-nucleosome complex (high gel mobility complex), the two linker DNAs bound to the BAF dimer are oriented perpendicularly. This arrangement is highly similar to the orientation of the two 7 base-pair DNA fragments bound to the BAF dimer in the previously reported BAF-DNA complex structure (Supplementary Fig. 3a; Bradley et al., NSMB 2005).

Additionally, in the low gel mobility complex, where two BAF dimers bridge two nucleosomes, the orientation of the linker and nucleosomal DNAs bound to the BAF dimers is also approximately perpendicular (Supplementary Fig. 3b). These observations suggest that BAF preferentially binds two DNAs with a perpendicular orientation, and may play a role in the formation of specific higher-order chromatin structures.

We added these descriptions to the revised manuscript in the following sections:

Lines 90-93:

“The two linker DNA fragments bound to the BAF dimer in the nucleosome are perpendicularly aligned, consistent with the orientation of the two DNA fragments in the BAF-DNA complex reported previously (Supplementary Fig. 3a)²⁸.”

Lines 135-139:

“The orientation of the linker and nucleosomal DNAs bound to an additional BAF dimer is nearly vertical, and similar to that of the linker DNAs bound to the BAF dimer in the high gel mobility complex (Supplementary Fig. 3b). The positions of SHL \pm 7.5 and SHL \pm 3 allow the two BAF dimers to bind symmetrically to the nucleosomal DNAs with the perpendicular alignment around the SHL \pm 7.5 and SHL \pm 3 regions.”

In both structures of the bridged nucleosomes, the BAF/Lamin complex binds to the same SHL locations. Are there structural constraints that would prevent BAF from binding to other locations, such as histone tails? Additionally, in both structures, BAF binds one DNA strand within the nucleosome particle (SHL \pm 3), while the other location is in the linker DNA. Is it structurally necessary for BAF to have at least one DNA in the linker region, not bound by histone proteins? Can BAF bridge two nucleosomes without any linker DNA?

Reply)

As described above, the SHL \pm 7.5 and SHL \pm 3 positions allow the symmetrical bridging of nucleosomes by two BAF dimers with the perpendicular alignment of the DNAs. This nucleosome binding mode may not be accomplished in the other SHL locations of nucleosomes. We described this fact in the revised text (Lines 135-139).

It would be interesting to know if the inner nuclear membrane protein emerin could bind to the BAF-nucleosome complex, given its role in tethering chromatin to the nuclear periphery/nuclear envelope. It should be straightforward for the authors to overlay their structure of the BAF-Lamin A/C complex bound to the nucleosome with the previously published structure of the ternary complex involving Lamin A/C, BAF, and emerin. Are there any structural clashes that would prevent emerin from binding to BAF in the structures described in this manuscript? In other words, are structural rearrangements of BAF necessary while bound to the nucleosome in order to bind emerin, or can emerin bind to the structures as presented in the manuscript? It would be interesting to address this point in the discussion.

Reply)

As this reviewer suggested, in the revised manuscript, we overlaid the previously published structure of the BAF-Lamin A/C IgF-emerin complex with the present BAF-Lamin A/C IgF-nucleosome complex structures (Supplementary Figure 5). We then found that emerin could bind to the BAF dimer in both BAF-Lamin A/C IgF-nucleosome complexes without steric clash. We present these new data in the new Supplementary Fig. 5, and discuss this fact in the Discussion section (lines 201-206).

The authors should improve the methods section: as it stands, it would be impossible for another scientist to reproduce their work. For example, it is unclear if the proteins can be frozen or not, and the same applies to the cryo-EM sample. The methods section lacks concentration details for any sample in all methods. For instance, in the nucleosome pull-down assay, there is no information on how much bead material was used, how much sample, their concentrations and volumes, or the ratio of nucleosomes to binding proteins; the same applies to other methods sections.

Reply)

Thank you for this comment. We added detailed descriptions to the Methods section regarding sample storage conditions. Additionally, we included sample concentrations, volumes, and ratios for the nucleosome pull-down assay as well as for the sample preparation for cryo-EM analysis.

Minor comments:

1. The authors sometimes write Lamin A/C Lys483, and other times Lamin A/C Lys486.

Reply)

Thank you very much. Lys483 was a typo. We corrected it in the revised manuscript.

2. In Supplementary Figure 5c, the number of particles for the initial 3D classification is missing.

Reply)

We added the number of particles for the initial 3D classification of the BAF-Lamin A/C IgF-H1-nucleosome complex in the corresponding Figure in the revised manuscript.

3. What is the unmapped gray density in Figure 3a (middle and right panels) between the DNA and the dark blue Lamin A/C?

Reply)

Thank you very much for this comment. The unmapped gray density should be the C-terminal region of the Lamin A/C IgF domain, corresponding to the unmodeled C-terminal 20 amino acid region of Lamin A/C (411-566). We corrected the gray density area to the color of Lamin A/C (blue) in the new Figure 3a.

4. The gel showing the quality of the BAF-Lamin A/C IgF-H1-nucleosome complex should be shown.

Reply)

We added a gel image showing the preparation of the BAF-Lamin A/C IgF-H1-nucleosome complex in Supplementary Figure 4a.

Reviewer #2 (Remarks to the Author):

The association between mitotic chromosomes and the nuclear envelope is a critical aspect of cell biology, yet its structural details remain largely unexplored. Previous studies have identified that Barrier-to-autointegration factor (BAF) anchors chromatin to the nuclear lamina via Lamin A/C. In this manuscript, the authors present a structural analysis of BAF and Lamin A/C binding to the nucleosome using cryo-EM. They discovered that the Lamin A/C IgF domain binds to the nucleosome dyad, while BAF interacts with both the Lamin A/C IgF domain and linker DNA. Additionally, they demonstrated that histone H1 can substitute for Lamin A/C. Notably, the study reveals that the BAF-Lamin A/C complex can bridge two nucleosomes or chromatosomes. Furthermore, mutations in the Lamin A/C residues Lys483 and His506, which are associated with lipodystrophy, were found to significantly impair nucleosome binding.

The structures elucidated in this study provide valuable insights into how BAF, Lamin A/C, and histone H1 associate with nucleosomes, which may have broader implications for chromatin organization within the nucleus. The findings are both novel and functionally significant.

Technical Comments:

1. Cross-Linking Concerns: A primary concern is the use of cross-linking in the cryo-EM study, which could potentially introduce artifacts. To address this, the authors should demonstrate that the nucleosome or chromatosome dimers exist in solution independently of cross-linking. Experiments such as gel filtration, analytical ultracentrifugation, small-angle light scattering, or generating a low-resolution cryo-EM density map without cross-linking would be beneficial.

Reply)

To address the concern regarding potential artifacts introduced by crosslinking, we performed ultracentrifugation in sucrose gradients with or without glutaraldehyde (Supplementary Figure 1e,f). The resulting gel images showed that the low and high mobility complexes are formed and separated by the sucrose-gradient ultracentrifugation under both conditions with and without glutaraldehyde, although the ratio of the low mobility complex is reduced under the latter conditions. These new results suggest that glutaraldehyde may stabilize the nucleosome bridging by BAF, but does not artificially induce the nucleosome bridging. We added the following description to the manuscript (lines 81-85).

“These high and low mobility complexes were then separated by sucrose gradient ultracentrifugation with glutaraldehyde fixation (GraFix) (Supplementary Fig. 1f). Two complexes were also observed in sucrose gradient ultracentrifugation without glutaraldehyde, although the ratio of the low mobility complex was reduced (Supplementary Fig. 1e,f). Therefore, these two complexes, with high and low mobilities, are not artificially formed due to crosslinking.”

2. Lamin A/C Full-Length Protein: The study utilized only the Lamin A/C IgF domain, not the full-length protein. Conducting binding assays with the full-length Lamin A/C, if it can be expressed and purified, would enhance the impact of the paper by providing a more comprehensive understanding.

Reply)

We appreciate the suggestion to include binding assays with the full-length Lamin A/C protein. We agree that isolating and studying full-length Lamin A/C, along with BAF, inner nuclear membrane proteins, and chromatin, would provide a more comprehensive understanding. However, we have not successfully purified the full-length Lamin A/C protein yet. Since it is a quite challenging project, we will save this important question for a future project.

3. Binding at SHL ± 3 and SHL ± 7.5 : The manuscript should provide an explanation for the observed binding at SHL ± 3 and SHL ± 7.5 positions.

Reply)

In the revised manuscript, we provided the following explanation for the observed binding at the (SHL) ± 7.5 and (SHL) ± 3 positions (Lines 135-139).

“The orientation of the linker and nucleosomal DNAs bound to an additional BAF dimer is nearly vertical, and similar to that of the linker DNAs bound to the BAF dimer in the high gel mobility complex (Supplementary Fig. 3b). The positions of SHL ± 7.5 and SHL ± 3 allow the two BAF dimers to bind symmetrically to the nucleosomal DNAs with the perpendicular alignment around the SHL ± 7.5 and SHL ± 3 regions.”

4. Clarification in Extended Data Figure 2: It is unclear whether Lamin A/C IgF interacts with H1 in Extended Data Figure 2. Clarification of this interaction is needed.

Reply)

To address this point, we added a close-up view of the region around Lamin A/C IgF and H1 in the BAF-Lamin A/C IgF-H1-nucleosome complex (new Supplementary Figure 4e,f). The distance between the main chains of the closest regions of Lamin A/C IgF and H1 is approximately 8Å, suggesting that the globular domain of H1 and the IgF domain of Lamin A/C do not interact directly. In the revised manuscript, we included the following description (lines 163-166).

“In the structure of the BAF-Lamin A/C IgF-H1-nucleosome complex, the distance between the nearest main chain regions of Lamin A/C IgF and H1 is approximately 8Å, indicating that the globular domain of H1 and the IgF domain of Lamin A/C may not directly interact (Supplementary Fig. 4f).”

5. Comparison with Chromatosome Structures: A comparison of the H1.1 chromatosome structure reported here with previously published high-resolution structures of chromatosomes containing either H1.0 or H1.4 (Zhou et al., Mol Cell, 2021; PMID: 33238161) would be useful. This could help determine whether the Lamin A/C IgF domain affects H1 binding during chromatosome dimer formation.

Reply)

Thank you very much for this insightful comment. To address this, we compared chromatosomes containing H1.0, H1.4, or H1.10 with the chromatosome in the BAF-Lamin A/C IgF-H1.1-nucleosome complex, as shown in the new Supplementary Figure 4d. The overall structure of the chromatosome in the BAF-Lamin A/C IgF-H1.1-nucleosome complex is quite similar to those of the chromatosomes containing H1.0,

H1.4, or H1.10, although a slight difference exists. Therefore, the H1 subtype may not affect the BAF-Lamin A/C binding to the chromosome. We described this fact in the revised manuscript (lines 155-159).

“In this study, we used H1.1, which reportedly co-exists with BAF, for the cryo-EM analysis. The overall structure of the chromosome in the BAF-Lamin A/C IgF-H1.1-nucleosome complex is quite similar to those of the chromosomes containing H1.0, H1.4, or H1.10, although a slight difference exists (Supplementary Fig. 4d). Therefore, the H1 subtype may not affect the BAF-Lamin A/C binding to the chromosome.”

Reviewer #3 (Remarks to the Author):

Barrier-to-autointegration factor (BAF) binds to mitotic chromosomes and upon mitosis exit assists the recruitment of lamins and other proteins for the assembly of both the lamina and the nuclear envelope and in anchoring chromatin to the nuclear envelope via the lamina. This is realized through specific interactions with lamin A/C. Yet, the underlying mechanism of this process remained obscure. The manuscript aims to solve the structure of the the BAF-Lamin A/C complex bound to nucleosomes and thus, to shed light on the lamina assembly and chromatin anchorage. The authors have initially solved the Cryo-EM structure of the BAF-Lamin A/C IgF-nucleosome complex by using highly purified recombinant proteins and reconstituted nucleosomes. The data showed that BAF-Lamin A/C binds the nucleosome by tripartite DNA binding and that the nucleosome binding of the lamin A/C IgF domain is dependant of the binding of the BAF dimer. This was followed by mutational analysis of nucleosome binding by BAF and lamin A/C and subsequent determination of the nucleosome complexes assembled with the mutated proteins. The results evidence that the K486N and H506D mutations of lamin A/C led to defective nucleosome binding. Since these mutations are observed in lipodystrophy patients, this could be associated with the aetiology of the disease. In addition, the authors showed that BAF-Lamin A/C complexes were able to connect two nucleosomes.

This is a nice manuscript which addresses an important problem and successfully solves it. From technical point of view, the manuscript is perfect. The experiments are well designed and professionally carried out. The text is easy to read and understand. I have only few minor notes.

1. Lanes 82-83: "...which may correspond to the high gel mobility complex...". This sentence should be rewritten. In order to claim this, the different complexes should be purified by glycerol/sucrose gradient and then separated on a native gel, which will identify the mobility of the complexes.

Reply)

Thank you very much for this comment. In the revised manuscript, we rewrote the corresponding sentences (ll.81-85).

2. Lanes 84, 86 : It would be better to remove "interestingly" and "surprisingly" from these lanes as well as throughout the text.

Reply)

We removed them accordingly.

3. Lane 87: "..Lys483 and His506..". Lys483 should be replaced with Lys486.

Reply)

Thank you for this comment. We corrected it accordingly.

4. lane 122, 123: idem point 1.

Reply)

Thank you very much for this comment. In the revised manuscript, we rewrote the corresponding sentence (ll.81-85).

5. lane 136: "...compacts the nucleosome structure 33,34. References 33&34 are not the most appropriate ones and should be substituted with references that reflect the claim "...forms tripartite binding interactions with the dyad and linker DNAs and compacts the nucleosome structure".

Reply)

Thank you for the suggestion. We replaced these references with those that describe the structures of chromosomes containing linker histone H1 subtypes.

6. lane 189: "will pave" should be replaced with "would pave".

Reply)

Thank you for the comment. We corrected it to "would pave".

7. lane 377, 378: the length of the linker between the biotin and the base used for the preparation of the 193 bp Widom sequence should be mentioned.

Reply)

Thank you for this comment. We added a more detailed description about the materials used in the pull-down assay.

8. lane 391, 392: "Nucleosomes, BAF, and Lamin A/C IgF, with or without H1.1, were mixed and incubated at 30°C for 30 min". The composition of the buffer used in the binding reaction has to be described. Are the nucleosomes containing already H1.1 or H1.1 is added together with BAF and Lamin A/C IgF? How the H1.1 bound nucleosomes are prepared?

Reply)

Thank you for this comment. We added more detailed descriptions of the experimental methods in the section on sample preparation for cryo-EM.